# Research on Fault Feature Extraction Method of Rolling Bearing Based on SSA–VMD–MCKD

**Zichang Liu, Siyu Li, Rongcai Wang and Xisheng Jia \***

Equipment Command and Management Department, Shijiazhuang Campus, Army Engineering University of PLA, Shijiazhuang 050003, China
**\*** Correspondence: xs_jia1964@163.com; Tel.: +86-155-3227-6287

**Abstract:** In response to the problem that nonlinear and non-stationary rolling bearing fault signals are easily disturbed by noise, which leads to the difficulty of fault feature extraction, to take full advantage of the superiority of variational mode decomposition (VMD) in noise reduction, and of maximum correlation kurtosis deconvolution (MCKD) in highlighting continuous pulses masked by noise, a method based on sparrow search algorithm (SSA), VMD, and MCKD is proposed, namely, SSA–VM–MCKD, for rolling bearing faint fault extraction. To improve the feature extraction effect, the method uses the inverse of the peak factor squared of the envelope spectrum as the fitness function, and the parameters to be determined in both algorithms are searched adaptively by SSA. Firstly, the parameter-optimized VMD is used to decompose the fault signal to obtain the intrinsic mode function (IMF) components, from which the optimal mode component is selected, and then the optimal component signal is deconvoluted by the parameter-optimized MCKD to enhance the periodic fault pulses in the optimal component signal, and finally extracts the rolling bearing fault characteristic frequency by envelope demodulation. Experiments on simulated signals and measured data show that the method can adaptively determine the parameters in VMD and MCKD, enhance the fault impact components in the signals, and effectively extract the fault characteristic frequencies of rolling bearings, with a success rate up to 100%, providing a new idea for rolling bearing fault feature extraction.

**Keywords:** rolling bearing; variational mode decomposition; maximum correlation kurtosis deconvolution; sparrow search algorithm; fault pulse; fault feature extraction

## 1. Introduction

Rolling bearings, as the main components of large industrial rotating machinery, play an important role in improving the efficiency of machinery and equipment, and their health condition directly affects the smooth operation of machinery and equipment [1]. However, rolling bearings often work in high temperature, high pressure, and a complex mechanical environment, and are highly susceptible to failure. Since the shock generated by the early failure of rolling bearings is very weak and easily disturbed by system noise, coupled with the complex vibration transmission path of rolling bearings, the extraction of rolling bearing fault characteristics is very difficult [2,3]. Therefore, it is of great significance to process the original vibration signals of rolling bearings and extract the fault characteristics accurately and effectively for the determination of their working conditions, the prevention of unexpected accidents, and the improvement of equipment operation efficiency [4].

The periodic force shocks generated by rolling bearings excite high-frequency resonance and convolution between the bearing system components, creating a bearing periodic shock signal [5]. To effectively deal with nonlinear and nonstationary rolling bearing fault signals and accurately extract fault features from the processed signals, domestic and foreign experts and scholars have proposed many effective methods. Among them, empirical mode decomposition (EMD), local mean decomposition (LMD), wavelet transform, etc.,

are currently popular signal processing methods which yield good results in processing rolling bearing fault signals [6]. However, the above methods still have shortcomings. Both EMD and LMD are recursive mode decomposition, which are greatly affected by sampling frequency, and have problems such as modal confusion and endpoint effects, making it difficult to guarantee the decomposition accuracy [7]. The wavelet transform is not an adaptive signal analysis method, it has no criteria for the optimal wavelet basis and requires a pre-selection of wavelet basis functions and decomposition layers [8]. To address the shortcomings of the above nonlinear signal processing methods, Dragomiretskiy et al. proposed a new non-recursive modal decomposition algorithm, namely VMD, in 2014 [9]. A fault feature extraction method combining MCKD and VMD is proposed in the literature [10], which first enhances the signal by MCKD, then decomposes it using VMD, and uses the kurtosis criterion to filter the modes with high fault information before noise reduction and reconstruction, and finally performs fault feature extraction by envelope demodulation. The maximum weighted kurtosis index is used as the objective function for fault feature extraction in the literature [11], and the parameters of VMD are sought by a grasshopper optimization algorithm. VMD is improved in the literature [12] to address the problem that the number of modal decompositions in this method needs to be set artificially. In terms of fault feature extraction, convolutional neural networks and other artificial intelligence-assisted algorithms are known to cope very well with different digital image recognition tasks. Therefore, feature extraction techniques producing two-dimensional digital images from scalar vibration signals have also been successfully exploited in bearing fault diagnosis. A rolling bearing fault identification method based on permutation entropy is proposed in the literature [13], using CNN for processing and classification of PE images. A fault identification method based on frequency domain downsampling and CNN is proposed in the literature [14], where the time-domain vibration signal is converted to the frequency domain by FFT, the sample processing is processed using maximum offset downsampling to obtain the network model input signal, and the features of the frequency-domain signal are extracted by CNN. In addition, deconvolution is a well-established method that can efficiently extract periodic pulses from fault signals [15]. Minimum entropy deconvolution (MED) seeks the optimal filter by maximizing the cragginess of the filtered signal, and extracts the fault characteristics of the signal by preserving the shock components of the signal [16]. In [17], the shock component in the signal is enhanced by MED. However, this method has the disadvantage that it is difficult to accurately distinguish between periodic pulses and random pulses. To avoid the problem that the MED can only highlight a few large tip pulses and can cause other shocks to be lost, McDonald et al. proposed the MCKD method in 2012, which is very suitable for extracting continuous transient shocks of weak fault signals by highlighting continuous shock pulses drowned by noise through deconvolution operations, improving the correlation kurtosis value of the original signal [18]. In the literature [19], the fault signal is decomposed into five IMF components using VMD and the fault frequency is extracted by MCKD, and the maximized correlation kurtosis is applied to enhance the periodic pulse components to achieve rolling bearing rolling body fault frequency extraction. However, this method has the disadvantage of poor adaptability by artificially setting parameters in VMD and MCKD. In the literature [20], SSA was used to find the parameters of MCKD, and then the bearing failure feature frequencies were extracted directly using the SSA-MCKD method. However, this method does not process the signal before the fault feature extraction. The literature [21] proposed PSO–VMD–MCKD for fault feature extraction. However, this method only performs a parameter search for the filter length and deconvolution period in MCKD, and the filter displacement number is artificially specified. Therefore, this method has the disadvantage of poor adaptability.

Although the existing research on fault feature extraction methods has made great progress, there are still some shortcomings in the existing methods:

(1) When the VMD and MCKD parameters are searched for, the existing methods have problems such as slow convergence and weak stability, reducing the computational efficiency of the entire process.

(2)　The existing fault feature extraction methods that combine VMD with MCKD, in which only some of the parameters are optimized, have the deficiency of poor self-adaptation.

In summary, this paper takes rolling bearings as the engineering research background and proposes a new rolling bearing fault feature extraction method, SSA–VMD–MCKD, to respond to the shortage of existing fault feature extraction methods, in order to give full play to the superiority of VMD in noise reduction and the advantages of MCKD in highlighting continuous pulses masked by noise, implementing adaptive selection of all parameters. This method effectively integrates the advantages of VMD in processing rolling bearing fault signals and the excellent ability of MCKD in extracting fault features. All the parameters of VMD and MCKD are adaptively optimized by the SSA optimization algorithm, which has the advantages of a wide application range, few adjustable parameters, and strong robustness. Finally, the feasibility and effectiveness of the method are verified using simulated signals and measured data, respectively.

The remaining sections of the paper are constructed as follows:

(1)　Section 2 first introduces the basic principles of VMD, MCKD and SSA to optimize the VMD and MCKD parameters.
(2)　Section 3 introduces the specific steps and flow chart of the SSA–VMD–MCKD method.
(3)　Section 4 validates the feasibility and effectiveness of the SSA–VMD–MCKD method for the extraction of rolling bearing fault features through simulation signals.
(4)　In the Section 5, based on the successful verification of the simulation signal, the CWRU and XJTU-SY data sets are used to further verify the feasibility and effectiveness of the SSA–VMD–MCKD method in practical engineering applications.
(5)　The conclusion is provided in Section 6.

## 2. SSA–VMD–MCKD Method

Firstly, the basic principles and processes of optimizing VMD and MCKD parameters by VMD, MCKD and SSA are introduced, respectively.

### 2.1. Principle and Process of VMD

By improving Wiener filtering and Hilbert transform, VMD applies the variational problem-solving process in signal decomposition, and solves the bandwidth and center frequency of the modal function. It is a new non-recursive modal decomposition algorithm. It can effectively highlight the fluctuation characteristics of the signal in each frequency-domain and has excellent signal processing performance for rolling bearing faults [22]. The VMD algorithm can be divided into two main parts: the first part is the construction of the variational problem, and the second part is the solution of the variational problem. The construction of the variational problem can be described by the following Equation (1):

$$
\begin{cases}
\min\limits_{\{u_k\},\{\omega_k\}} \left\{ \sum\limits_{k=1}^{K} \left\| \partial_t \left[ \left( \delta(t) + \frac{j}{\pi t} \right) * u_k(t) \right] e^{-j\omega_k t} \right\|_2^2 \right\} \\
s.t. \sum\limits_{k=1}^{K} u_k(t) = f(t)
\end{cases}
\tag{1}
$$

where $\{u_k\} = \{u_1, u_2, \ldots, u_K\}$ represents the $K$ IMF components decomposed by the input signal $f(t)$: $u_k(t) = A_k(t) \cos[\varphi_k(t)]$; $A_k(t)$ and $\varphi_k(t)$ are the instantaneous amplitude and phase angle of $u_k(t)$, respectively; $\{\omega_k\} = \{\omega_1, \omega_2, \ldots, \omega_K\}$ is the center frequency of the IMF set, $\omega_k(t) = \mathrm{d}\varphi_k(t)/\mathrm{d}t$; $K$ is the number of decompositions; $\partial_t[\cdot]$ represents the partial derivative of the time parameter $t$; $\delta(t)$ is the impulse function; j is the imaginary unit; "*" indicates convolution; and the Hilbert-transformed one-sided spectrum of the modal components is $\left( \delta(t) + \frac{j}{\pi t} \right) * u_k(t)$ [23].

The second part is the solution of the variational problem. Here, a Lagrange multiplier $\lambda$ that can increase the strictness of constraints and a quadratic penalty factor $\alpha$ that can reduce

interference from Gaussian noise are introduced. The constrained variational problem in Equation (1) is turned into an unconstrained variational problem, and then solved [24]:

$$L(\{u_k\}, \{\omega_k\}, \lambda) = \alpha \sum_{k=1}^{K} \left\| \partial_t \left[ \left( \delta(t) + \frac{j}{\pi t} \right) * u_k(t) \right] e^{-j\omega_k t} \right\|_2^2 +$$
$$\left\| f(t) - \sum_{k=1}^{K} u_k(t) \right\|_2^2 + \left\langle \lambda(t), f(t) - \sum_{k=1}^{K} u_k(t) \right\rangle \tag{2}$$

By alternating the direction multiplier algorithm, one can iteratively update $u_k^{n+1}$, $\omega_k^{n+1}$, and $\lambda^{n+1}$ to obtain the "saddle point", that is, the optimal solution of Equation (1). The specific process is [25]:

Step 1: initialize parameters $\hat{u}_k^1$, $\omega_k^1$, $\hat{\lambda}^1$, $\alpha$, and $n$;

Step 2: $k = k + 1$, until $k = K$, update $u_k$ and $\omega_k$ according to Formula (3):

$$\begin{cases} \hat{u}_k^{n+1}(\omega) = \dfrac{\hat{f}(\omega) - \sum\limits_{i<k} \hat{u}_i^{n+1}(\omega) - \sum\limits_{i>k} \hat{u}_i^n(\omega) + \frac{\hat{\lambda}(\omega)}{2}}{1 + 2\alpha(\omega - \omega_k^n)^2} \\ \omega_k^{n+1} = \dfrac{\int_0^\infty \omega \left| \hat{u}_k^{n+1}(\omega) \right|^2 \mathrm{d}\omega}{\int_0^\infty \left| \hat{u}_k^{n+1}(\omega) \right|^2 \mathrm{d}\omega} \end{cases} \tag{3}$$

where Fourier transform is performed on $u_k$ and $\lambda$; $\hat{u}_k$ and $\hat{\lambda}$ can be obtained;

Step 3: update $\lambda$:

$$\hat{\lambda}^{n+1}(\omega) = \hat{\lambda}^n(\omega) + \tau \left( \hat{f}(\omega) - \sum_k \hat{u}_k^{n+1}(\omega) \right) \tag{4}$$

where $\tau$ is the noise tolerance parameter.

Step 4: determine whether the conditions for stopping the iteration are met by Formula (5). If so, stop the iteration; if not, return to step 2:

$$\sum_k \left( \left\| \hat{u}_k^{n+1} - \hat{u}_k^n \right\|_2^2 / \left\| \hat{u}_k^n \right\|_2^2 \right) < \varepsilon \tag{5}$$

where $\varepsilon > 0$ is the given discrimination accuracy, which is usually set to $1 \times 10^{-6}$.

### 2.2. Principle and Process of MCKD

The MCKD method aims at maximizing the correlation kurtosis and extracts the periodic impulse components of the signal by iteratively selecting a suitable filter for deconvolution operation. The correlation kurtosis expression is [26]:

$$CK_M(T) = \frac{\sum\limits_{n=1}^{N} \left( \prod\limits_{m=0}^{M} y_{n-mT} \right)^2}{\left( \sum\limits_{n=1}^{N} y_n^2 \right)^{M+1}} \tag{6}$$

where $M$ is the displacement number; $T = f_s / f_{fault}$ is the impulse signal period; $f_s$ and $f_{fault}$ are the sampling frequency and fault characteristic frequency, respectively; $m \in [0, M]$, the number of pulses becomes more with the increase of $M$, thus improving the fault feature extraction capability of the method, but too large an $M$ leads to a decrease in computational accuracy; the periodic impulse signal $y_n (n = 1, 2, \ldots, N)$ is the output signal, that is, the processed signal; $N$ is the length of the signal, and the expression of $y_n$ is [27]:

$$y_n = f * x = \sum_{l=1}^{L} f_l x_{n-l+1} \tag{7}$$

where $L$ is the filter length; $f = [f_1, f_2, \ldots, f_L]$ is the filter coefficient; $l \in [0, L]$; and $x_n$ is the input signal.

After filtering the signal, the objective function of MCKD is shown in Formula (8), which maximizes the correlation kurtosis:

$$
\begin{aligned}
\text{MCKD}_M(T) &= \max_f CK_M(T) \\
&= \max_f \frac{\sum\limits_{n=1}^{N}\left(\prod\limits_{m=0}^{M} y_{n-mT}\right)^2}{\left(\sum\limits_{n=1}^{N} y_n^2\right)^{M+1}}
\end{aligned} \tag{8}
$$

The maximum value of $CK_M(T)$ in Equation (8) can be obtained by solving Equation (9):

$$
\frac{\mathrm{d}}{\mathrm{d}f_l} CK_M(T) = 0 \tag{9}
$$

The filter coefficient expression in matrix form is further obtained as:

$$
f = \frac{\|y\|^2}{2\beta^2}\left(X_0 X_0^{\mathrm{T}}\right)^{-1} \sum_{m=0}^{M} X_{mT} \psi_m \tag{10}
$$

where

$$
\beta = \begin{bmatrix} y_1 y_{1-T} \cdots y_{1-MT} \\ y_2 y_{2-T} \cdots y_{2-MT} \\ \vdots \\ y_N y_{N-T} \cdots y_{N-MT} \end{bmatrix} N \times 1;
$$

$$
X_r = \begin{bmatrix} x_{1-r} & x_{2-r} & x_{3-r} & \cdots & x_{N-r} \\ 0 & x_{1-r} & x_{2-r} & \cdots & x_{N-1-r} \\ 0 & 0 & x_{1-r} & \cdots & x_{N-2-r} \\ \vdots & \vdots & \vdots & \ddots & \vdots \\ 0 & 0 & 0 & \cdots & x_{N-L-r+1} \end{bmatrix} L \times N;
$$
$$
(r = 0, T, \ldots, MT)
$$

$$
\psi_m = \begin{bmatrix} y_{1-mT}^{-1}\left(y_1^2 y_{1-T}^2 \cdots y_{1-MT}^2\right) \\ y_{2-mT}^{-1}\left(y_2^2 y_{2-T}^2 \cdots y_{2-MT}^2\right) \\ \vdots \\ y_{N-mT}^{-1}\left(y_N^2 y_{N-T}^2 \cdots y_{N-MT}^2\right) \end{bmatrix} N \times 1.
$$

In summary, the specific steps of MCKD are as follows [28]:

Step 1: determine parameters such as $L$, $T$, and $M$;

Step 2: calculate $X_0 X_0^{\mathrm{T}}$ and $X_{mT}$ of $x_n$;

Step 3: calculate the filtered signal $y_n$;

Step 4: calculate $\beta$ and $\psi_m$ according to $y_n$;

Step 5: update the filter coefficient $f$;

Step 6: if $\Delta CK_M(T) < \varepsilon$ before and after filtering, $\varepsilon$ is the set threshold, then stop the iteration; otherwise return to Step 3.

### 2.3. SSA Optimizes Parameters of VMD and MCKD

The performance of VMD and MCKD methods is greatly affected by the parameter values. Among them, the performance of the VMD method is strongly influenced by parameters $\alpha$ and $K$: when $\alpha$ increases, the bandwidth of each IMF component becomes smaller and the attenuation becomes faster. On the contrary, when $\alpha$ decreases, the bandwidth

becomes larger and the attenuation becomes slower. Too large a setting of $K$ will lead to over-decomposition, too small will lead to incomplete decomposition, leading the fault information to be ignored. The performance of the MCKD method is affected by $L$, $T$, and $M$: if $L$ is set too small, the deconvolution accuracy will decrease, and conversely, the number of samples to be calculated in each iteration will increase, thereby prolonging the calculation time. The setting of the parameter $T$ requires an a priori fault period; the fault pulses of the deconvoluted signal increase as the parameter $M$ becomes larger [29]. If the above parameters are set subjectively by humans, the effect of fault feature extraction will deteriorate. Therefore, an optimization algorithm is required to adaptively determine each parameter in the above two methods.

Xue, J.K. proposed an optimization algorithm in 2020, namely, SSA. Compared with the algorithms such as the genetic algorithm and the ant colony algorithm, SSA has a better optimization effect, and has the advantages of high accuracy, strong robustness, fast convergence speed, and wide application range [30]. Therefore, to obtain better fault feature extraction, the parameters of VMD and MCKD are optimally sought using SSA to adaptively obtain the required parameter combinations.

In SSA, it is divided into three parts: explorer, follower, and early warning. The explorer is mainly used to determine the foraging search direction and area. The position update can be expressed as [31]:

$$X_{i,j}^{t+1} = \begin{cases} X_{i,j}^t \cdot \exp\left(\frac{-i}{\alpha \cdot M}\right) & if \ R_2 < ST \\ X_{i,j}^t + Q \cdot \boldsymbol{L} \ if \ R_2 \geq ST \end{cases} \tag{11}$$

where $X_{i,j}^t$ denotes the position of the $i$-th sparrow in the $j$-th dimension at the $t$-th iteration; $\exp(\cdot)$ represents the exponential function with the natural constant $e$ as the base; $\alpha \in [0, \ 1]$ is a random number; $M$ is the maximum number of iterations; $R_2 \in [0, \ 1]$ is the warning value; $ST \in [0.5, \ 1]$ is the safety value; $Q$ is a random number obeying normal distribution; $\boldsymbol{L}$ is a $1 \times d$-dimensional matrix, with all elements being 1. If $R_2 < ST$, it means that the explorer can continue to expand the search range. If $R_2 \geq ST$, it indicates that the explorer needs to leave the current area and turn to a safe location to forage.

The follower's position update Formula is [32]:

$$X_{i,j}^{t+1} = \begin{cases} Q \cdot \exp\left(\frac{X_{\text{worst}}^t - X_{i,j}^t}{i^2}\right) if \ i > N/2 \\ X_P^{t+1} + \left|X_{i,j}^t - X_P^{t+1}\right| \cdot \boldsymbol{A}^+ \cdot \boldsymbol{L} \ otherwise \end{cases} \tag{12}$$

where $X_{\text{worst}}$ is the worst position on the board; $X_P$ is the optimal position of the explorer; $A$ is a $1 \times d$-dimensional matrix with each element randomly assigned 1 or $-1$, and $\boldsymbol{A}^+ = \boldsymbol{A}^{\text{T}}\left(\boldsymbol{A}\boldsymbol{A}^{\text{T}}\right)^{-1}$. If $i > N/2$, it indicates the need to go to other areas for forage. Otherwise, it means that the follower has the chance to forage.

The early-warning person is responsible for guarding against predation, and the location update Formula is [33]:

$$X_{i,j}^{t+1} = \begin{cases} X_{\text{best}}^t + \beta \cdot \left|X_{i,j}^t - X_{\text{best}}^t\right| \ if \ f_i \neq f_g \\ X_{i,j}^t + K \cdot \left(\frac{\left|X_{i,j}^t - X_{\text{worst}}^t\right|}{(f_i - f_w) + \varepsilon}\right) \ if \ f_i = f_g \end{cases} \tag{13}$$

Among them, $X_{\text{best}}$ indicates the best position; $\beta$ is a random number used to control the step size; $K \in [-1, \ 1]$ is a random number used to control the step size and moving direction; $f_i$, $f_w$, and $f_g$ denote the individual fitness, the worst fitness, and the optimal fitness, respectively.

When using SSA to optimize VMD and MCKD parameters, the fitness function is the inverse of the peak factor squared of the envelope spectrum. The expression of the peak factor is:

$$Ec = \frac{\max(X(z))}{\sqrt{\sum_z X(z)^2 / Z}} \tag{14}$$

where $X(z)(z = 1, 2, \ldots, Z)$ is the signal envelope spectrum amplitude sequence. The larger the crest factor, the more obvious the fault characteristic. Therefore, the smaller the value of the fitness function, the stronger the periodic impact characteristic and the better the optimization effect.

## 3. Fault Feature Extraction Process

To effectively integrate the strengths of the VMD method for processing rolling bearing fault signals, the superiority of the MCKD method for extracting fault features, and to adaptively seek all parameters in the VMD and MCKD methods, a new rolling bearing fault feature extraction method, namely SSA–VMD–MCKD, is proposed. The SSA–VMD–MCKD method flow is shown in Figure 1.

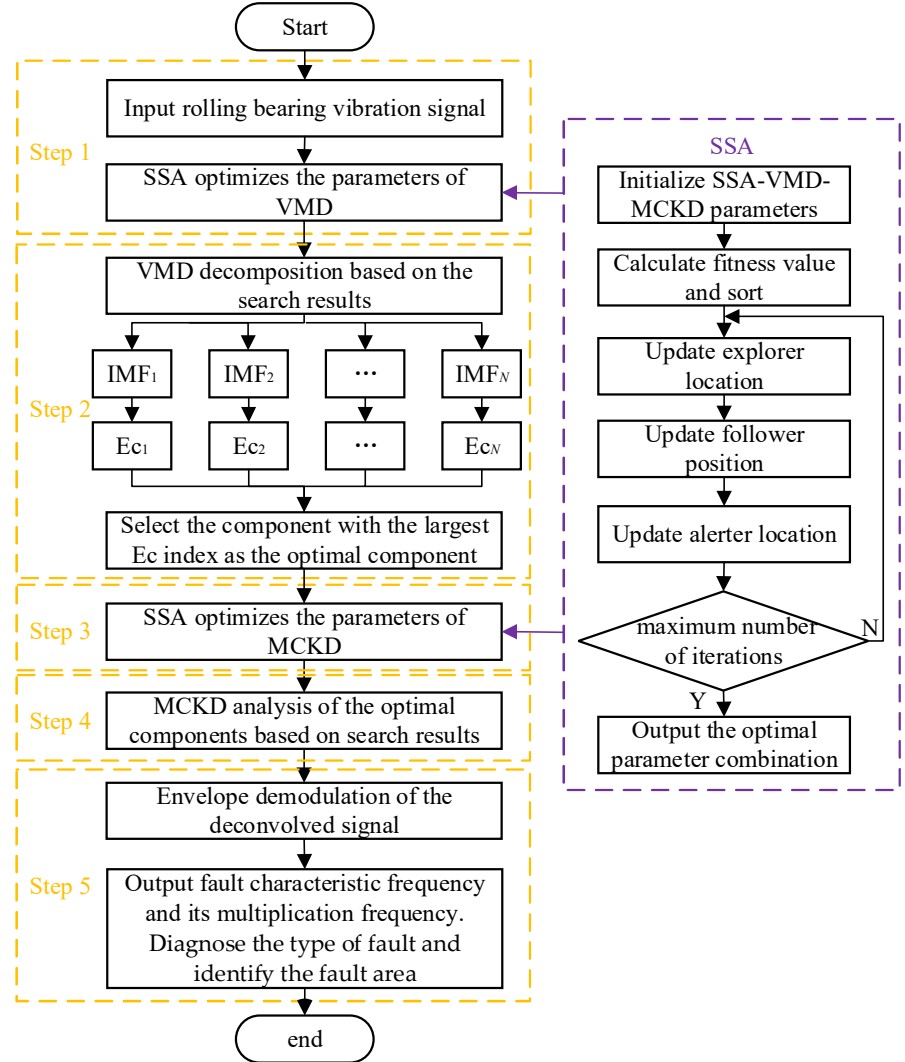

**Figure 1.** Flow chart of the SSA–VMD–MCKD method.

According to the flow chart of the SSA–VMD–MCKD method, the specific steps are as follows:

Step 1: acquisition of raw vibration signals using sensors on rolling bearings. According to the vibration signal, use SSA to perform adaptive optimization on the parameters of VMD and obtain the required combination $[\alpha_0, K_0]$. According to the literature [34] and combined with the actual situation, the number of sparrow populations in SSA is set to 30, the maximum number of iterations is 20, the proportion of explorers is 0.2, the safety value is 0.8, the optimization range of parameter $\alpha$ is [100, 2000], and the optimization range of parameter $K$ is [3, 10].

Step 2: set the parameters in VMD according o the optimal parameter combination obtained in step 1. Perform VMD processing of the fault signal to obtain the required components of each IMF. Then, calculate the peak factor value of the envelope spectrum of each component according to Formula (14) and select the component with the largest value as the required optimal component. Envelope demodulation analysis is performed on the optimal component, the frequency band containing the frequency line with the largest amplitude in the envelope spectrum is the prominent frequency range, and the frequency band is a symmetric interval with the prominent spectral line as the symmetry axis.

Step 3: according to the optimal components obtained in step 2, use SSA to perform adaptive optimization on the parameters of MCKD, and obtain the required parameter combination $[L_0, T_0, M_0]$ of filter length, impulse signal period, and shift number. The parameter settings in SSA are the same as in step 1. According to the literature [35] and combined with the actual situation, set the optimization range of parameter $L$ to [100, 1000]. After comprehensively considering the accuracy of fault feature extraction and computational efficiency, the optimization range of parameter $T$ is selected as $[f_s / f_{\text{fault}} - 30, \ f_s / f_{\text{fault}} + 30]$ and the optimization range of parameter $M$ is [3, 7]. When the fault characteristic frequency is unknown, the frequency band highlighted from step 2 is used to determine the range of the $T$ search in the MCKD parameters. The size of the frequency band should be moderate. The maximum value of the upper boundary is slightly larger than the maximum fault characteristic frequency of the theoretical bearing in the whole transmission system.

Step 4: after setting the parameters in MCKD according to the optimal parameter combination obtained in step 3, use the MCKD method to analyze the optimal component to strengthen the periodic fault pulses in the signal.

Step 5: perform envelope demodulation on the deconvolved signal and obtain the fault characteristic frequency value and its multiplication frequency. The theoretical fault characteristic frequency values of the bearings are compared with the spectral lines with significant peaks in the envelope spectrum to diagnose the fault type.

Through the above steps, effective extraction of rolling bearing fault characteristics and determination of fault types can be achieved.

## 4. Simulation Signal Analysis

When measuring the data, due to the influence of factors such as the accuracy of the sensor and the operating environment (temperature, humidity), the collected vibration signal usually contains a certain degree of error. To verify the feasibility and effectiveness of the method proposed in the paper, the impact signal generated by the bearing inner ring failure is simulated using the rolling bearing failure model, and strong Gaussian white noise is added to simulate the faint failure of the actual working condition of the bearing covered by ambient noise. The original simulation signal is shown in Equation (15) [36]:

$$\begin{cases} x(t) = y(t) + n(t) = \sum_{k=1}^{K} A_k h(t - kT - \tau_k) + n(t) \\ A_k = A_0 \sin(2\pi f_r t) + 1 \\ h(t) = \exp(-Ct) \sin(2\pi f_n t) \end{cases} \tag{15}$$

where attenuation coefficient $C = 800$; resonance frequency $f_n = 4000$ Hz; amplitude $A_0 = 0.5$; rotation frequency $f_r = 25$ Hz; sampling frequency $f_s = 12,800$ Hz; fault characteristic frequency $f_{\text{fault}} = 120$ Hz; the number of analysis points is 8192; $\tau_k$ means obeying a small fluctuation with a mean of zero and a standard deviation of $0.5\% f_r$; $n(t)$ represents the Gaussian white noise component; and the signal-to-noise ratio of the noise-containing signal is set to $-16$ dB.

The time-domain waveforms of the original signal and of the simulated signal after adding noise are shown in Figure 2.

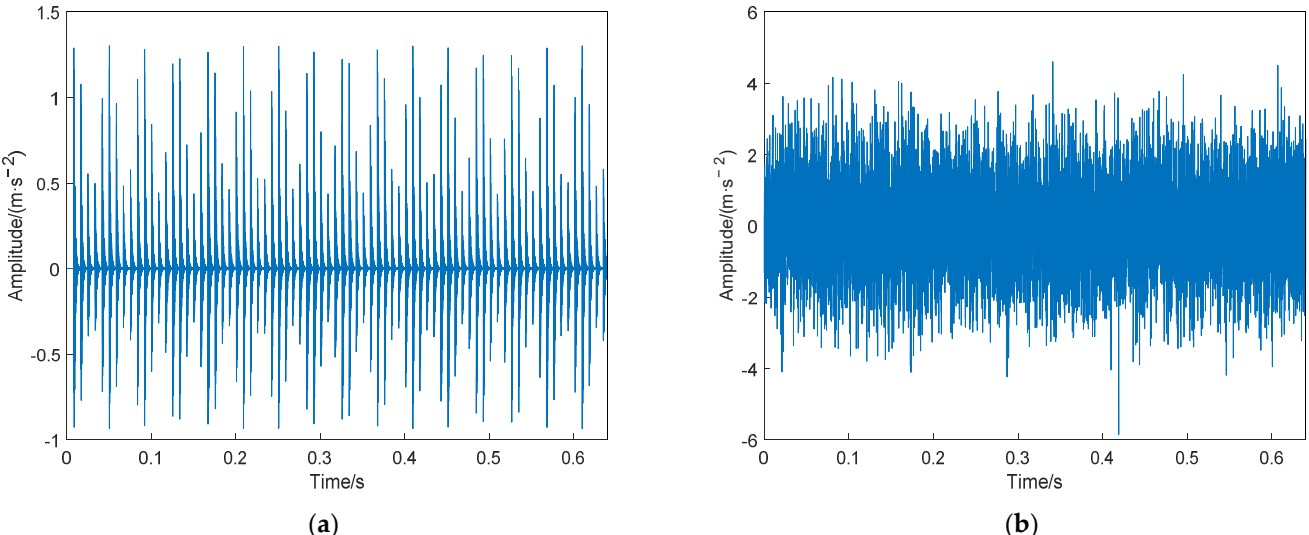

(**a**)

(**b**)

**Figure 2.** Time-domain waveforms of the original and noisy signals: (**a**) the original signal's time-domain waveform; (**b**) the time-domain waveform with a noisy signal.

As can be seen from Figure 2, the original signal contains obvious periodic shock components. After adding Gaussian white noise to the original signal, the periodic shock signal in Figure 2a is completely drowned out by the noise, and the original periodic impact component cannot be distinguished.

The frequency-domain waveform and envelope spectrum of the noisy signal are shown in Figure 3.

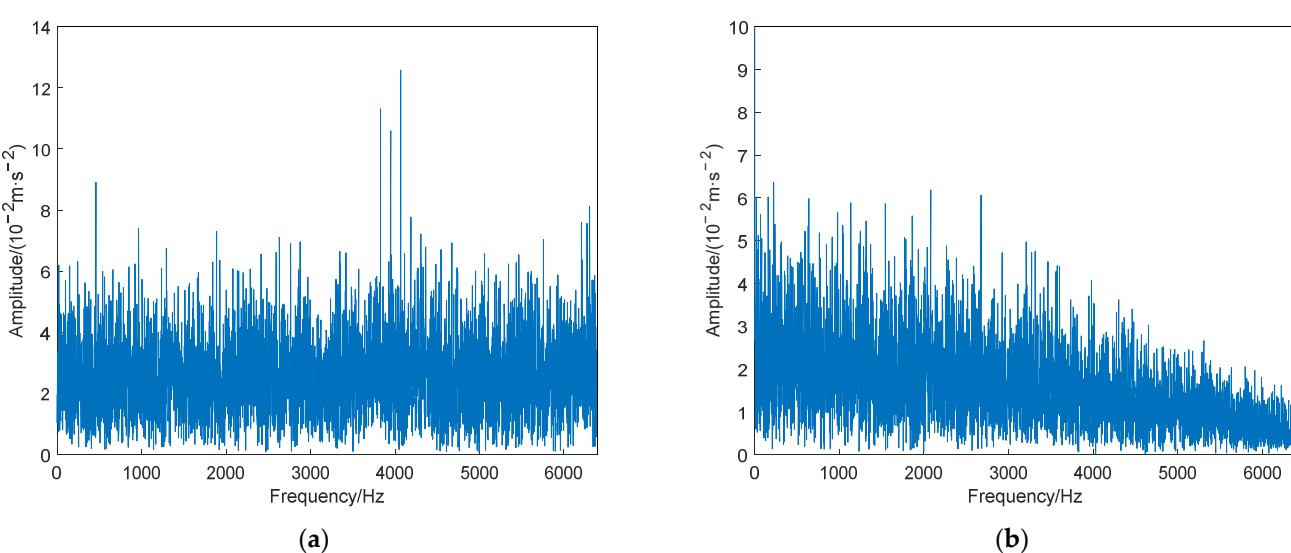

(**a**)

(**b**)

**Figure 3.** Frequency-domain waveform and envelope spectrum of the noisy signal: (**a**) frequency-domain waveform with noisy signal; (**b**) Envelope spectrum of noisy signal.

From Figure 3a, it can be concluded that the spectrum of the noise-containing signal has no regularity and the frequency components are disordered, making it difficult to find fault information. From the envelope spectrum of Figure 3b, it is difficult to find the prominent frequencies, and it is impossible to identify the fault information effectively. In summary, the traditional spectrum analysis method has failed.

The fault features are extracted using the SSA-VMD-MCKD method proposed in this paper. The curve of the variation of the value of the fitness function with the number of iterations is obtained by adaptive optimization seeking of the parameters of VMD through SSA as shown in Figure 4.

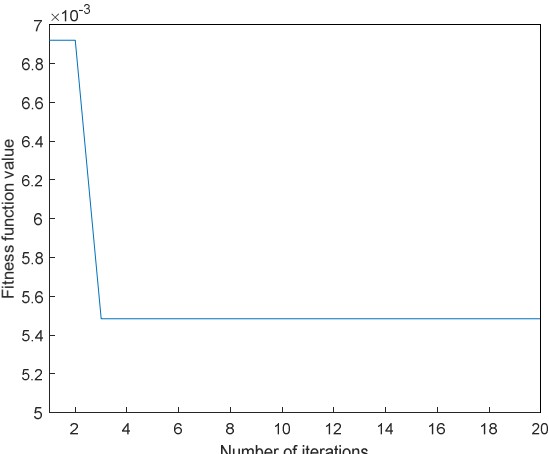

**Figure 4.** The fitness function varies with the number of iterations.

The SSA-optimized VMD converges in the third generation and the value of the fitness function searched is $5.5 \times 10^{-3}$, which is a dimensionless index. It can be concluded that the optimal combination of parameters $[\alpha_0, K_0]$ for VMD are $\alpha_0 = 1423$ and $K_0 = 9$, respectively. After setting the parameters in VMD according to the results obtained from the optimization search, the IMF components obtained by using VMD to process the signal containing noise are shown in Figure 5.

As can be seen from Figure 5, the VMD algorithm is used to decompose the simulated signal to obtain a total of nine IMF components, with the center frequencies of IMF1~IMF9 gradually increasing. From the waveform and frequency change of IMF components, it can be inferred that the forward component is the real signal component, and the higher frequency at the back may be the high-frequency noise component in the signal. The envelope spectral peak factor values of each IMF component in the above figure are calculated according to Equation (14), and the results obtained are shown in Figure 6.

The component with the largest value of the peak factor of the envelope spectrum is selected as the desired optimal component. The magnitude plot of each component in Figure 6 shows that the value of IMF4 is the largest by comparison; therefore, IMF4 is selected as the optimal component. Using the envelope spectrum to analyze this component, the results obtained are shown in Figure 7.

It can be observed that the amplitude of the corresponding spectral line is largest when the horizontal coordinate is taken as 120 Hz, and it can be inferred that this frequency is the most likely frequency close to the fault characteristics. However, in the bearing fault signal envelope spectrum, only $f_{\text{fault}}$ and its multiples can be observed to ensure effective extraction of fault features, and the 120 Hz multiples component does not appear in the above figure; therefore, it cannot be determined that 120 Hz is the fault frequency. In order to accurately extract the fault characteristics, further analysis should be performed. According to the prominent spectral line with the largest amplitude in the above figure, the frequency band containing 120 Hz is selected as the prominent frequency range, and the frequency band [90, 150] is selected as the frequency range for solving T after comprehensive consideration.

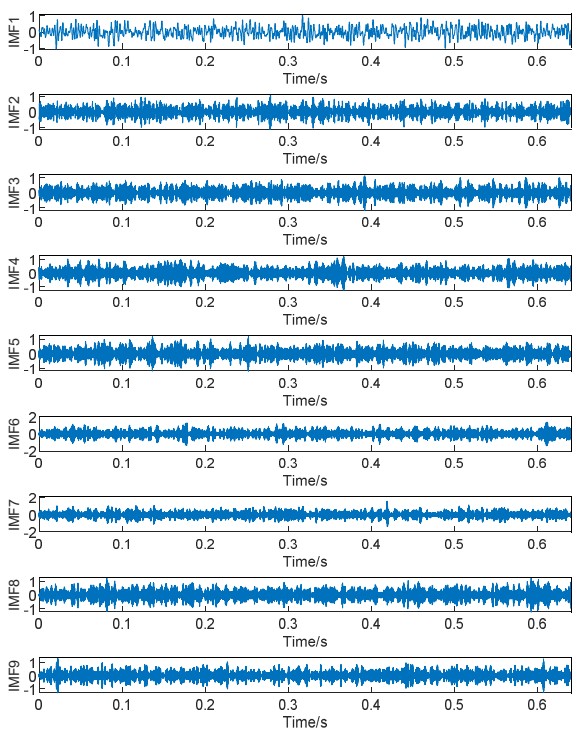

**Figure 5.** VMD decomposition results.

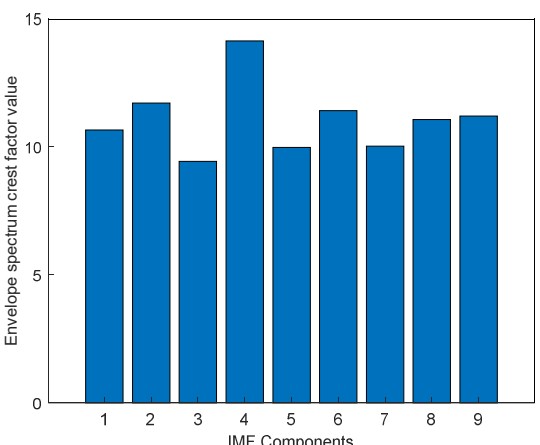

**Figure 6.** Each component's envelope spectrum crest factor value.

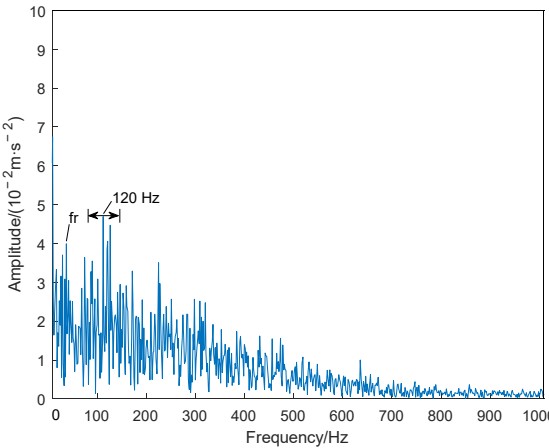

**Figure 7.** The envelope spectrum of the optimal component.

To highlight the superiority of parameter-optimized VMD, the noise-containing signal is processed by CEEMDAN to obtain 15 IMF components. The peak factor value of the envelope spectrum of each component is calculated separately, the IMF12 with the largest index is taken as the optimal component for which the envelope demodulation analysis is performed, and the results obtained are shown in Figure 8.

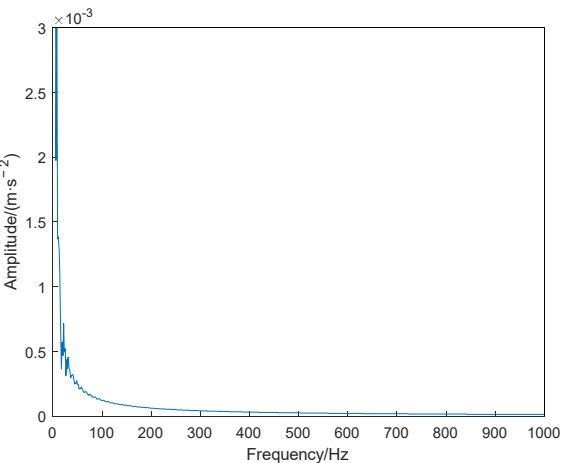

**Figure 8.** CEEMDAN optimal component envelope spectrum.

As can be seen from Figure 8, the highlighted frequency range does not include the fault characteristic frequencies. Therefore, the superiority of using SSA to optimize VMD parameters is verified.

The parameters of MCKD are adaptively searched for using SSA Figure 9 shows the curve of the value of the fitness function of the optimized MCKD algorithm with the number of iterations of population evolution; the vertical coordinates are dimensionless indicators.

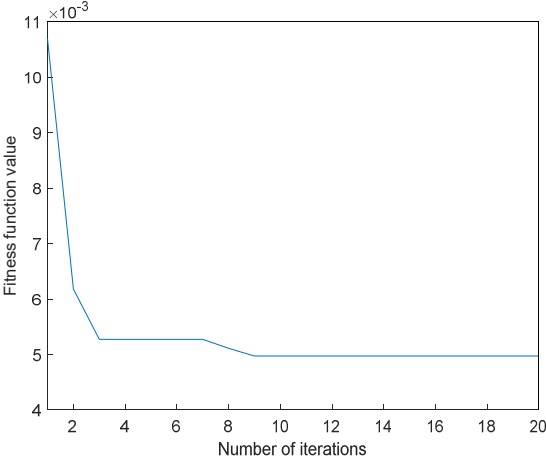

**Figure 9.** The fitness function varies with the number of iterations.

From the above figure, it can be seen that SSA converges in the ninth generation and the fitness function takes the minimum value $5 \times 10^{-3}$. The optimal combination of parameters is obtained as $L_0 = 740$, $T_0 = 99$ and $M_0 = 5$. After setting the parameters in MCKD according to the optimal parameter combination obtained, use MCKD to analyze the optimal component, and obtain the processed time-domain plot and envelope spectrum, as shown in Figure 10.

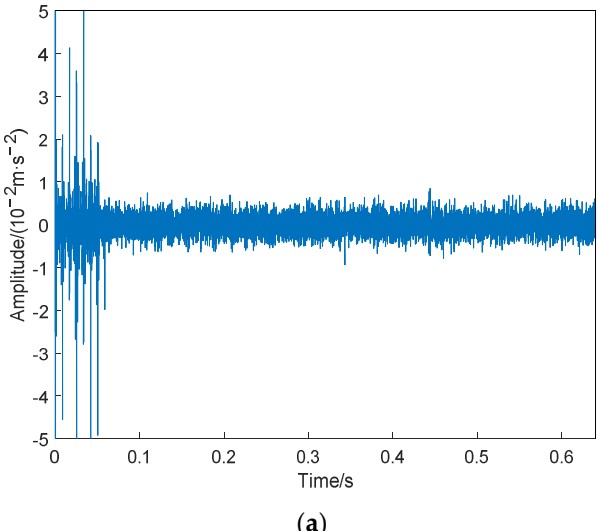
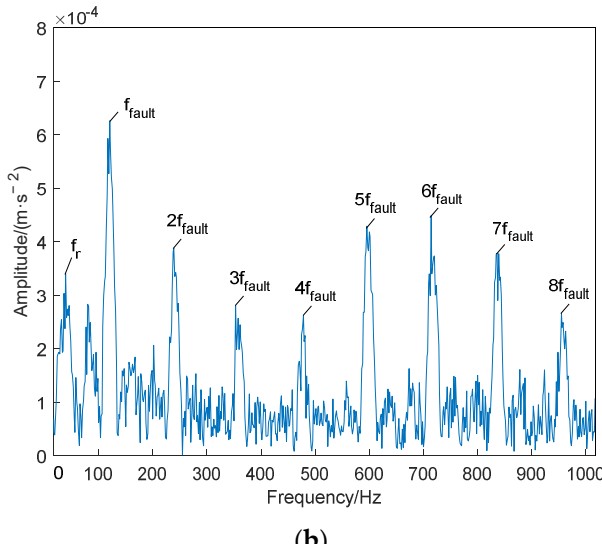

(**a**)　　　　　　　　　　　　　　　　　　　　　　　　　(**b**)

**Figure 10.** Time-domain waveform and envelope spectrum after MCKD processing: (**a**) time-domain graph after MCKD processing; (**b**) envelope spectrum after MCKD processing.

The optimal component is processed by MCKD to highlight continuous shock pulses that are drowned by noise and to improve the correlation kurtosis value of the original signal. The extraction of fault features can be accomplished by envelope demodulation. After completing the above steps, the fault shock component is clearly observable, and the signal shock component is significantly increased in the time-domain waveform of Figure 10a. The spectral lines of both the fault characteristic frequency and its multiples are clearly visible in the deconvoluted envelope spectrum of Figure 10b, and the fault type can thus be identified as an inner-loop fault, indicating that the characteristic frequency is accurately extracted, thus verifying the correctness of the present method.

The reliability of the results obtained by using SSA to optimize the MCKD parameters was verified by modifying the three parameters of MCKD. These were changed in accordance with 10% float from the original parameter value, and the filter length, impulse signal period, and shift number were changed to 666, 90, and 4, respectively. The results are shown in Figure 11.

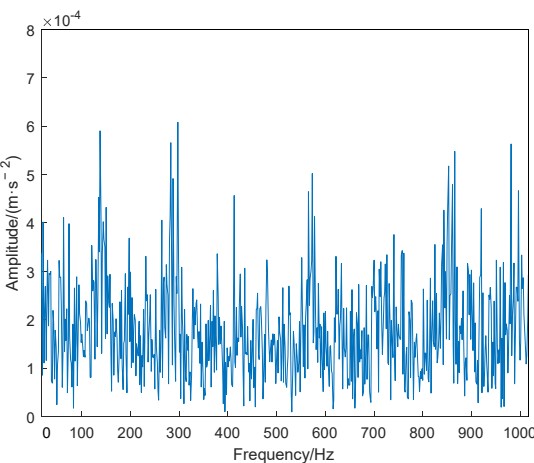

**Figure 11.** The envelope spectrum after modifying the parameters for MCKD processing.

It is more difficult to observe the fault characteristic frequency in Figure 11. There are many interferences spectral lines around it and the multiplication frequency is not obvious enough, which makes it impossible to extract the fault characteristics effectively, indicating that the results of optimizing MCKD parameters using SSA in this paper are reliable.

To verify the necessity of combining the optimized VMD algorithm and the optimized MCKD algorithm proposed in this paper, the simulated signal is analyzed without the optimized VMD preprocessing, based on the SSA optimized MCKD for the simulated signal. After optimization, the filter length, impulse signal period, and shift number are 848, 107, and 4, respectively, and the results obtained are shown in Figure 12.

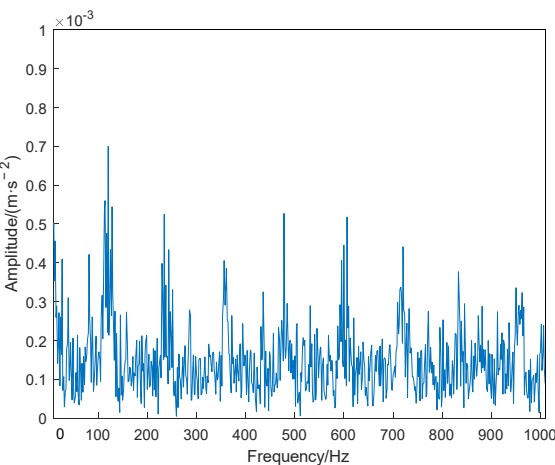

**Figure 12.** Envelope spectrum after direct MCKD processing.

Obviously, the direct use of MCKD with parameters optimized by SSA is not satisfactory for processing the rolling bearing fault signal, and there are more interference components in the envelope spectrum, which makes it difficult to clearly distinguish the fault feature frequencies and their multiples, resulting in difficulty in extracting the fault features. This proves the effectiveness of the fault feature extraction method proposed in this paper, and the necessity of using the parameter-seeking VMD to process the signal before MCKD analysis.

## 5. Measured Signal Analysis

Based on the successful verification of the simulated signals, the feasibility and effectiveness of the SSA–VMD–MCKD rolling bearing fault feature extraction method in practical engineering applications was further verified using the CWRU data set and the XJTU–SY data set [37].

### 5.1. Analysis of Vibration Data of CWRU Rolling Bearings

The effectiveness of applying the SSA–VMD–MCKD method to rolling bearing fault feature extraction was further validated using a publicly available bearing vibration signal data set from CWRU. The experiment uses deep groove ball bearings, model SKF6205. The failure of the bearing is single point damage with EDM, and the vibration acceleration signal of the bearing is collected using an acceleration sensor [38]. The specific data used is the inner ring fault vibration signal with motor load of 0, approximate speed of 1797 r/min, fault diameter of 0.1778 mm, rotation frequency $f_r = 1797/60 = 29.95$ Hz, and sampling frequency $f_s = 12{,}000$ Hz. According to the parameters provided by the data set, the inner ring fault characteristic frequency obtained is $f_i = 5.4152 f_r = 162.1852$ Hz [39].

The first 4096 sampling points of the data set are selected as the fault signal test data; the time-domain waveform and envelope spectrum of the experimental signal are shown in Figure 13.

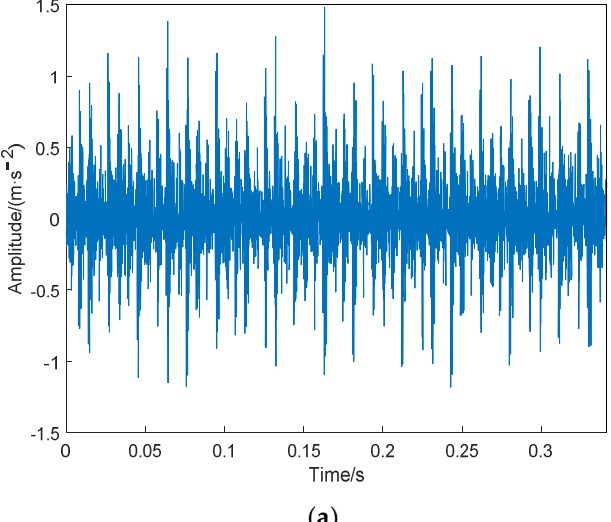
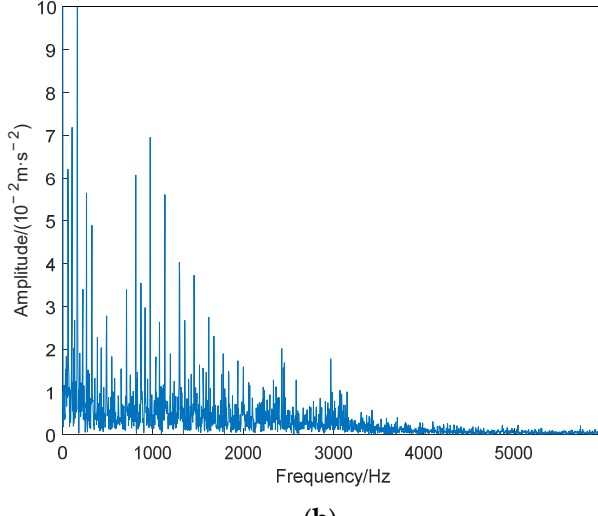

(**a**)
(**b**)

**Figure 13.** Time-domain waveform and envelope spectrum of the fault signal: (**a**) fault signal time-domain waveform; (**b**) Envelope spectrum of the fault signal.

Insignificant periodic shock components can be observed in the time-domain waveform of the fault signal, which is due to the unavoidable noise interference during data acquisition. Some of the shock components have been covered by noise. In addition, further envelope demodulation analysis of the signal, as shown in Figure 13b, did not reveal prominent frequency components.

The SSA–VMD–MCKD method was used to analyze the experimental signals. First, the parameters of VMD are adaptively optimized by SSA. Figure 14 shows the curve of the change in the value of the fitness function with the number of iterations of population evolution.

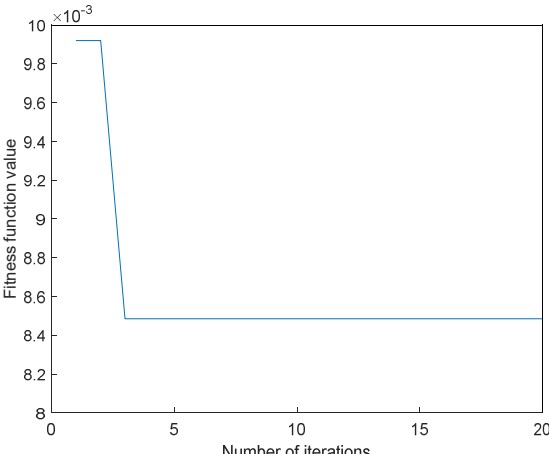

**Figure 14.** The fitness function varies with the number of iterations.

The SSA optimization converges in the third generation, the minimum value of the fitness function is taken to be $8.5 \times 10^{-3}$, and the optimal combination of parameters obtained is $\alpha_0 = 644$ and $K_0 = 8$, respectively. After setting the parameters in VMD according to the combination of the parameters obtained from the optimization search, VMD is performed on the fault signal and eight IMF components are obtained, as shown in Figure 15.

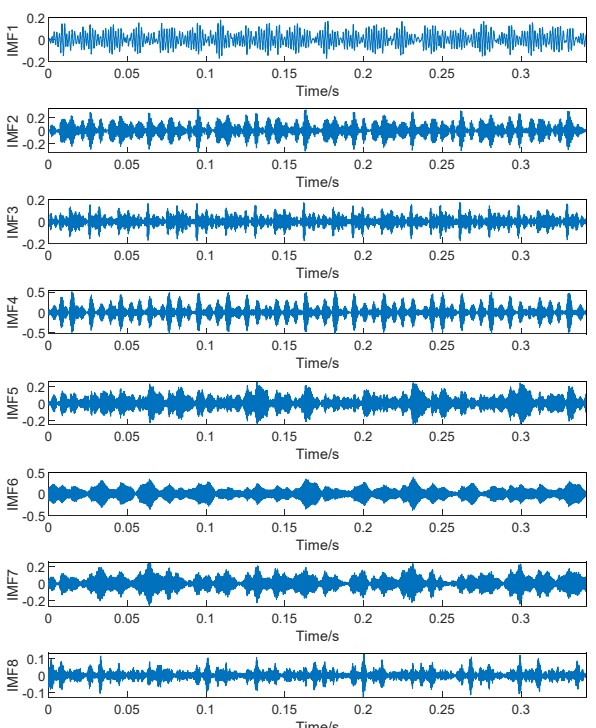

**Figure 15.** VMD decomposition result of fault signal.

The value of the envelope spectral peak factor is calculated for each IMF component in the above figure, and the result obtained is shown in Figure 16.

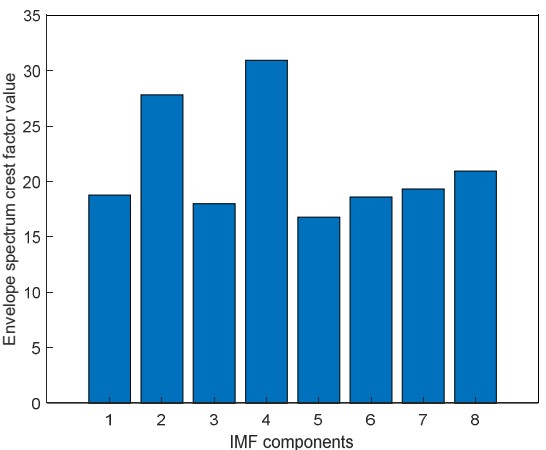

**Figure 16.** Each component's envelope spectrum crest factor value.

Through the plot of each component amplitude of each component in Figure 16, it can be seen that IMF4 has the largest value of the envelope spectrum peak factor; therefore, IMF4 is selected as the optimal component. Envelope spectrum analysis is performed for IMF4, and the results are shown in Figure 17.

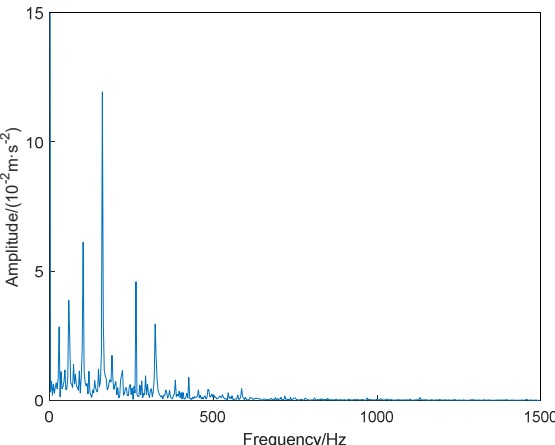

**Figure 17.** The envelope spectrum of the optimal component.

The 162.1852 Hz multiplier frequency component does not appear in the graph; thus, it is not possible to determine that 162.1852 Hz is the fault frequency. The frequency band [44, 103] was chosen as the frequency range for solving T. Next, use SSA to adaptively optimize the parameters of MCKD, and obtain the required filter length, impulse signal period, and shift number parameter combinations as $L_0 = 575$, $T_0 = 78$, and $M_0 = 4$. The curve of the change in the value of the fitness function with the number of iterations of population evolution is shown in Figure 18.

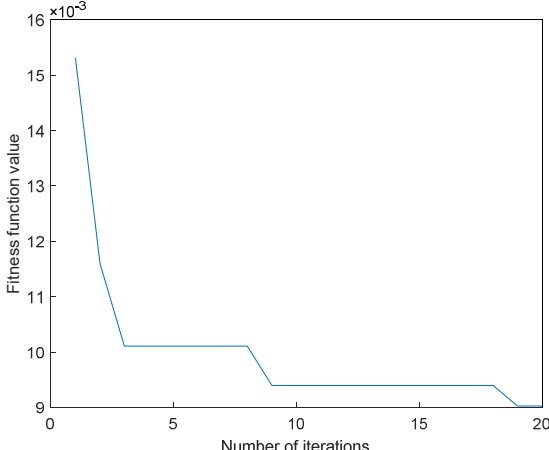

**Figure 18.** The fitness function varies with the number of iterations.

It can be seen from Figure 18 that when the number of iterations reaches 19, the fitness function takes the minimum value $9 \times 10^{-3}$. After setting the parameters in MCKD according to the optimal parameter combination obtained, use MCKD to perform deconvolution analysis on the optimal components and obtain the analyzed time-domain waveform and envelope spectrum, as shown in Figure 19.

From the time-domain waveform in Figure 19a, the shock component can be clearly observed. The spectral lines of the fault characteristic frequency and its multiples are clearly visible in the deconvoluted envelope spectrum, and the fault type of the fault signal can thus be identified as an inner-loop fault, indicating that the characteristic frequency is accurately extracted, thus verifying the effectiveness of the method proposed in this paper.

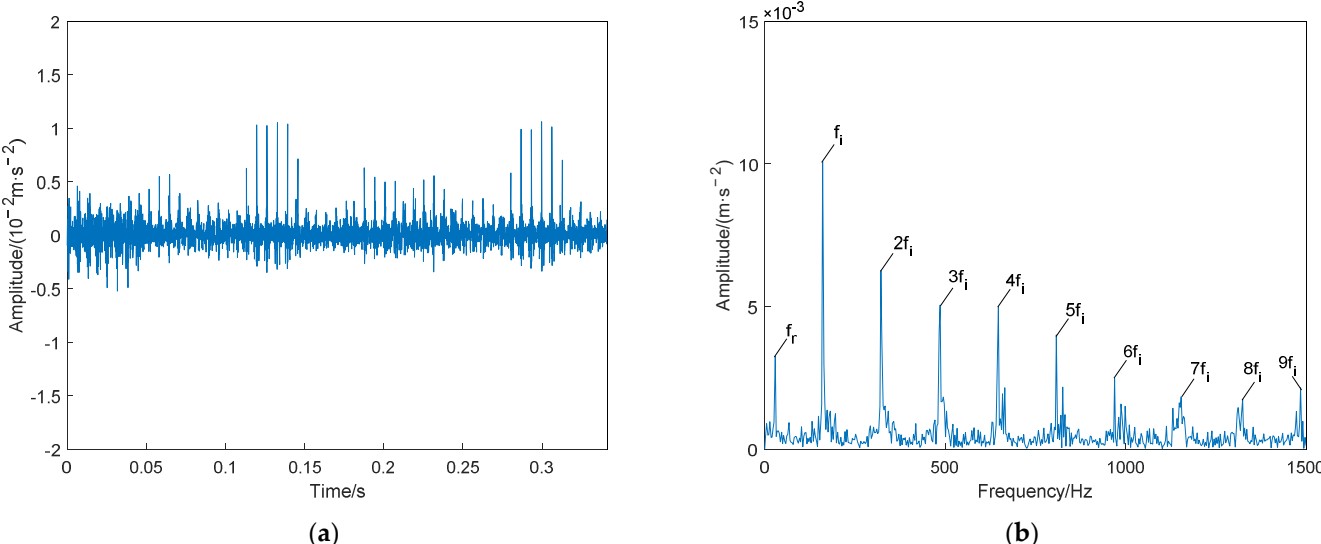

**Figure 19.** Time-domain waveform and envelope spectrum after MCKD processing: (**a**) time-domain graph after MCKD processing; (**b**) envelope spectrum after MCKD processing.

To verify the necessity of the method proposed in this paper to process the fault signal using VMD before MCKD analysis, the fault signal is directly analyzed using MCKD with parameters optimized by SSA. The results obtained are shown in Figure 20.

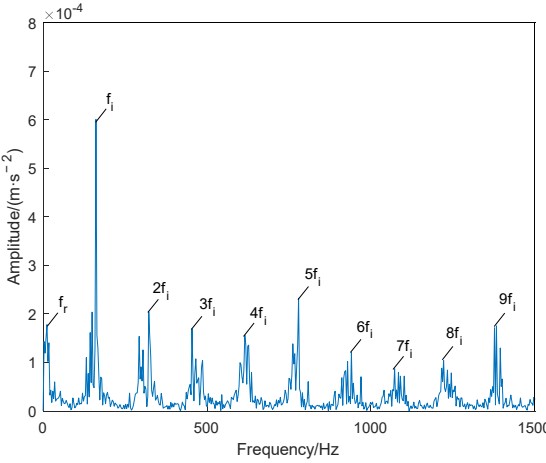

**Figure 20.** Envelope spectrum after direct SSA–MCKD processing.

Although the fault signal can be processed directly using MCKD with parameters optimized by SSA, and the fault feature frequencies and their multipliers can be obtained, the results are obviously not as clear and those obtained using the method proposed in this paper. This demonstrates the effectiveness of the fault feature extraction method proposed in this paper and the necessity of using VMD after parameter optimization to process the signal before MCKD analysis.

### 5.2. XJTU–SY Bearing Life Cycle Data Analysis

The XJTU–SY bearing life cycle data compiled by Dr. Wang Biao of Lei Yaguo's research group at Xi'an Jiao tong University are used to further verify the effectiveness of the method proposed in this paper [40]. The bearing accelerated life test platform is shown in Figure 21. The test bearing is LDK UER204, the number of balls is eight, and the contact angle is $0°$. An acceleration sensor is used to obtain the bearing vibration signal. The rotating speed is 2100 r/min, sampling frequency $f_s = 25,600$ Hz, rotation frequency $f_r = 2100/60 = 35$ Hz,

radial force is 12 kN, outer ring fault characteristic frequency $f_o = 107.91$ Hz, and a fault signal of a length of 4096 was selected for the analysis [41].

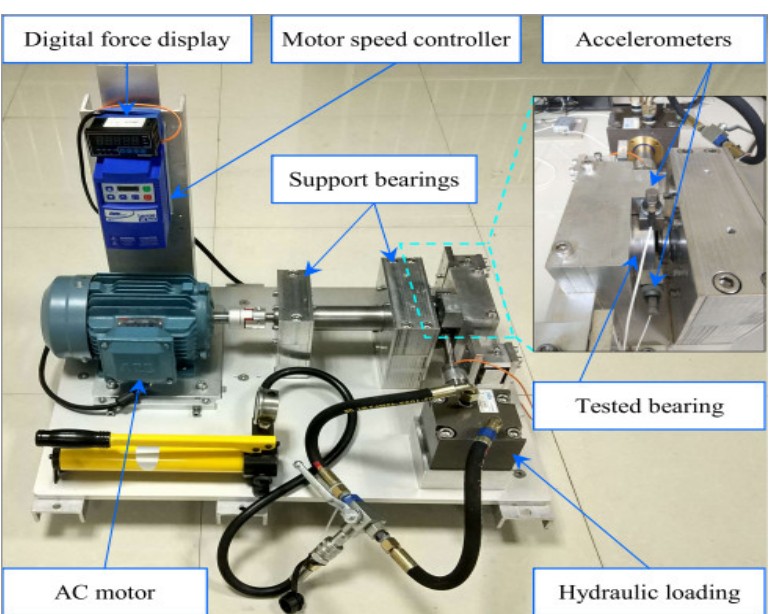

**Figure 21.** Bearing accelerated life test bench.

The time-domain waveform and envelope spectrum of the experimental signal are shown in Figure 22.

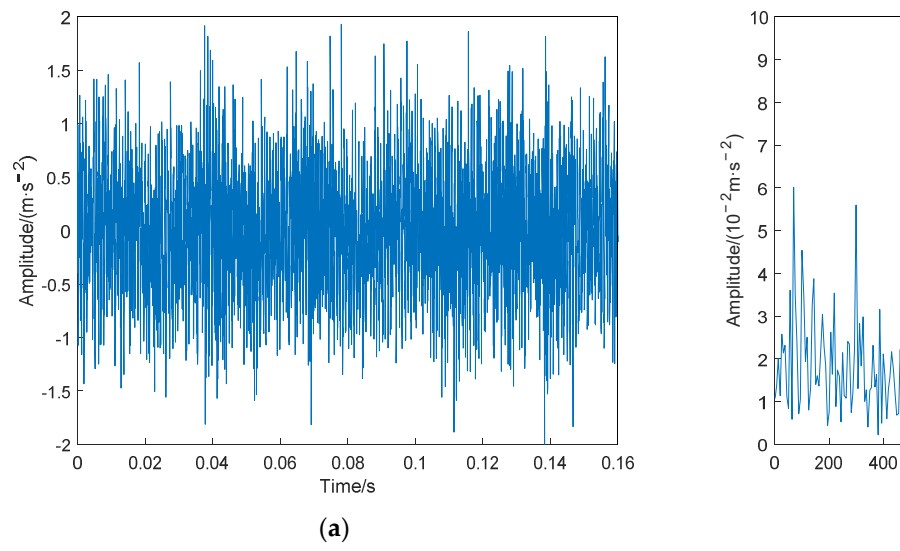

(**a**)

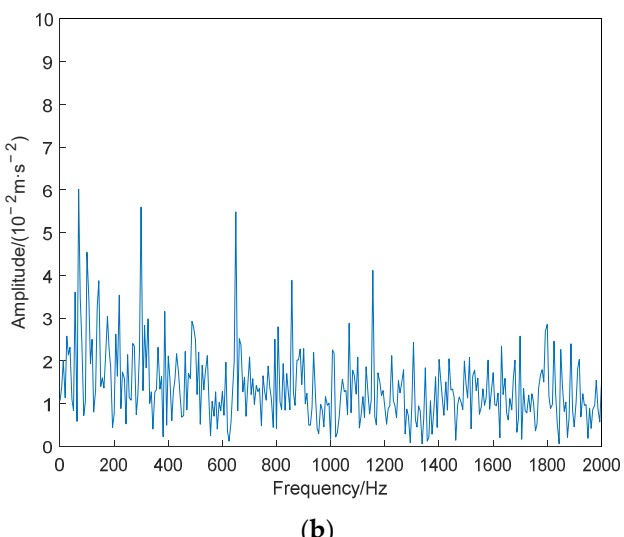

(**b**)

**Figure 22.** Time-domain waveform and envelope spectrum of fault signal: (**a**) fault signal time-domain waveform; (**b**) envelope spectrum of the fault signal.

The time-domain waveform hardly observes any shock component, and there is no pattern after the envelope demodulation analysis of the signal.

The experimental signal was analyzed using the method proposed in this paper. The parameters of VMD were optimized by SSA, and the variation curve of the fitness function value with the number of iterations is shown in Figure 23.

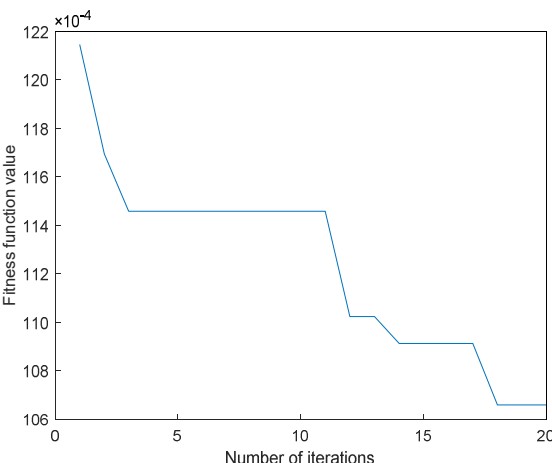

**Figure 23.** The fitness function varies with the number of iterations.

After obtaining the optimal parameter combinations of the number of components and the penalty factor in the VMD method as $K_0 = 8$ and $\alpha_0 = 904$, respectively, perform VMD processing on the fault signal, and obtain eight IMF components, as shown in Figure 24.

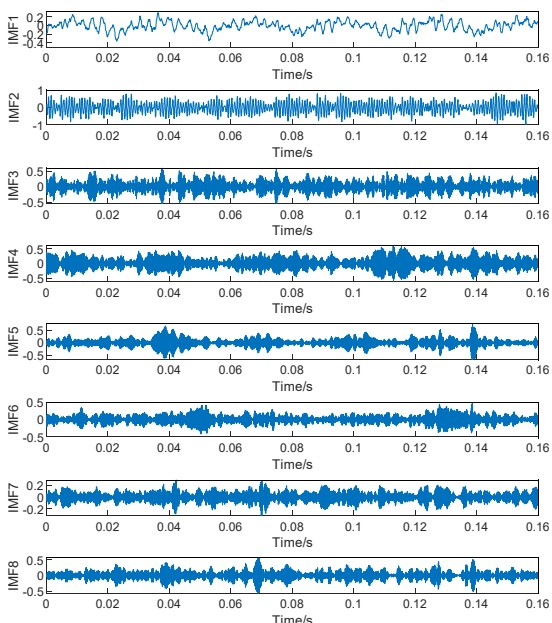

**Figure 24.** VMD decomposition result of fault signal.

Calculate the peak factor value of the envelope spectrum of each IMF component in the above figure; the result is shown in Figure 25.

It can be seen that IMF1 is the optimal component. In addition, the range of MCKD optimization parameters T is calculated as [207, 267]. SSA is used to adaptively optimize the parameters of MCKD, and the required filter length, impulse signal period, and shift number parameters are combined as $L_0 = 1111$, $T_0 = 229$, and $M_0 = 4$. The curve of the fitness function value changing with the number of iterations is shown in Figure 26.

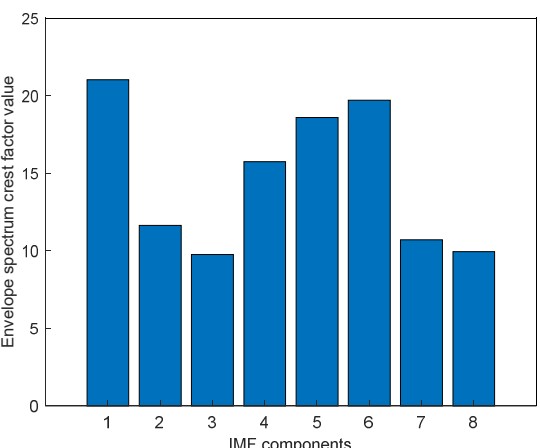

**Figure 25.** Each component envelope spectrum crest factor value.

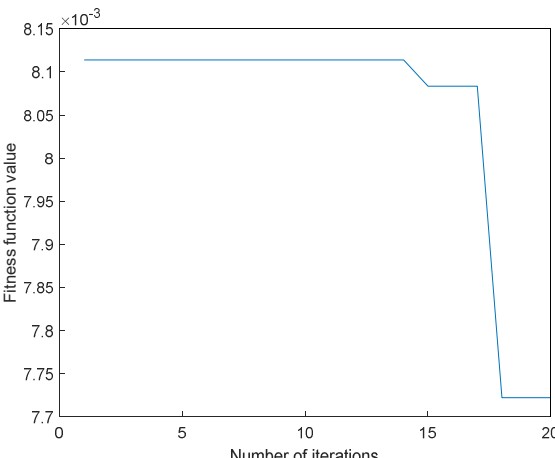

**Figure 26.** The fitness function varies with the number of iterations.

After setting the parameters in MCKD according to the optimal parameter combination obtained, the time-domain waveform and envelope spectrum obtained by deconvolution analysis of the optimal components, using MCKD, are shown in Figure 27.

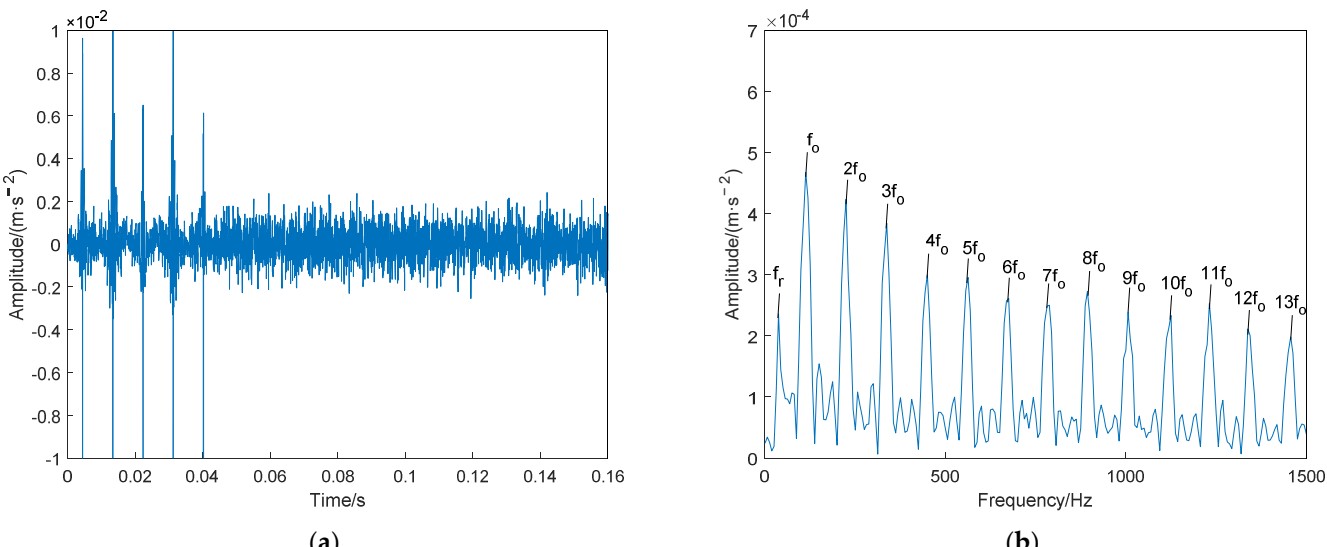

**Figure 27.** Time-domain waveform and envelope spectrum after MCKD processing: (**a**) time-domain graph after MCKD processing; (**b**) envelope spectrum after MCKD processing.

In Figure 27a, the signal impact component is significantly increased. In Figure 27b, the spectral lines of the fault feature frequency and its multiples are clearly visible, indicating that the rolling bearing fault features are effectively extracted, and the fault type of the fault signal can be determined as outer ring fault; thus, the effectiveness of the fault feature extraction method proposed in this paper is further verified.

Figure 28 shows the results of the outer ring fault signal by fast spectral kurtosis (FSK) processing [41].

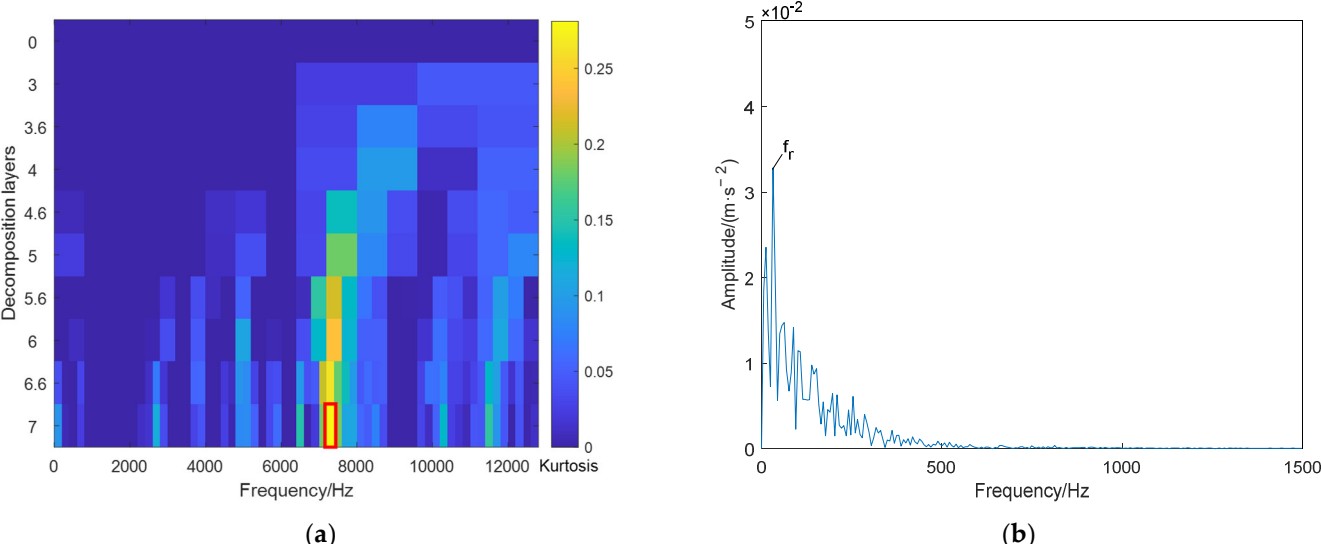

(**a**)　　　　　　　　　　　　　　　　　　　(**b**)

**Figure 28.** Fault signal FSK analysis results: (**a**) spectral kurtosis plot; (**b**) filtered envelope spectrogram.

The bandwidth of the feature signal with the largest kurtosis is 200 Hz and the center frequency is 7300 Hz. The bandwidth is narrow, the noise interference is large, only the transient frequency is found in the filtered signal envelope spectrum, and the fault feature frequency and its multiples cannot be observed, which is not effective.

## 6. Conclusions

This paper takes rolling bearings as the engineering research background and conducts a study on their fault feature extraction methods. Because the existing fault feature extraction methods usually have poor self-adaptation and an unsatisfactory fault feature extraction effect, a more suitable fault feature extraction method for rolling bearings is proposed: SSA–VMD–MCKD. Simulated signals and measured data verify the effectiveness of the proposed method. The following conclusions can be drawn:

(1) The SSA-based improved VMD method is capable of adaptively searching to obtain the decomposition modal number and penalty factor, avoiding the interference of human subjective factors on the selection of VMD parameters, achieving effective suppression of noise components, and highlighting transient shocks.

(2) The improved MCKD method based on SSA can adaptively search for the optimal combination of filter length, deconvolution signal period, and shift number. The interference of human subjective factors on the selection of MCKD parameters is avoided, and the optimal deconvolution of fault signals is achieved.

(3) The results of fault feature extraction from simulated signals and complex real measurement data show that it is difficult to accurately extract fault features under strong background noise by using only the optimized VMD method, or directly by using the optimized MCKD. On the other hand, the SSA–VMD–MCKD method can accurately extract the weak fault features of rolling bearings under strong background noise.

The SSA–VMD–MCKD method proposed in this paper can provide theoretical and technical support for rolling bearing fault features extraction and has good prospects for engineering applications.

The study of the fault feature extraction method proposed in this paper is only validated with the simulation data and the preset fault data of the test bench, and its generality needs to be further studied in conjunction with the actual equipment. Meanwhile, although the method proposed in this paper has a better fault feature extraction effect compared with other methods, there is a deficiency of long parameter search time, and more advanced intelligent optimization algorithms can be used to improve the method proposed in this paper.

**Author Contributions:** Data curation, Z.L. and S.L.; resources, S.L. and R.W.; supervision, X.J.; validation, R.W. and X.J.; writing—original draft, Z.L.; writing—review and editing, Z.L. All authors have read and agreed to the published version of the manuscript.

**Funding:** This work was funded by National Natural Science Foundation of China (Grant No. 71871219).

**Data Availability Statement:** These data were collected through many experiments and are copyrighted by the laboratory and therefore cannot be made public, thank you.

**Conflicts of Interest:** The authors declare no conflict of interest.

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
