# Peer review of "Research on Fault Feature Extraction Method of Rolling Bearing Based on SSA–VMD–MCKD"

_electronics, doi:10.3390/electronics11203404_

Round 1

Reviewer 1 Report

1. Abstract Section should be focused.

2. Motivation Section should be listed.

3. Flow chart of SSA-VMD-MCKD method should be discussed in-detail.

4. Nonparametric plots for the discussed data should be sketched.

5. Quotation marks should be followed.

6. More details about Figure 5 should be listed.

7. Time-domain waveform and envelope spectrum after MCKD processing; Time domain graph after MCKD processing; and Envelope spectrum after MCKD processing. Did the authors see a significant result?. Explain.  

8. Did the authors discussed the normality property of data?. Explain.

9. Conclusion Section should be rewritten. 

Reviewer 2 Report

The paper seems to have little novelty. However, if the authors seriously improve the paper, it can be published.  The authors have to address the following major points:

1. English language needs some improvement throughout the paper. Please check and correct the grammatical mistakes in the whole document. Please, there are many other errors in this document in several places, please carefully refine the English language.

2. The document really needs a general nomenclature which must be placed at the beginning before the introduction section grouping all the symbols and characters used in the work in addition to the all the abbreviations.

3. You should add quantitative results in the abstract section.
4. You should add appropriate references for all equations.

5. The introduction is long and requires significant revision. it is therefore recommended to condense or sift this section properly, focusing on what is necessary while reducing the sentences and at the same time deleting redundant or repetitive expressions
6. The Introduction is not comprehensive in terms of exploring the literature review.
7. The exact novelty of the work in relation with others should be highlighted in the Introduction.
8. The advantages of the methodology proposed by the authors to extract the modeling should be present clearly.
9. The flowchart needs to redrawing to show the inputs and outputs in addition to the processes and decisions that used in the analysis.
10. It should be listed the assumptions (as points), and what are their implications on the results.
11. The authors must provide a greater discussion of the results with interpretations in physical meaning.
12. The authors should think over the real significance of their results and try to rewrite this section to improve understanding of the conclusions

13.  Please extend conclusions. I suggest adding a paragraph with limitations of the discussed approach as well as future research directions.

Reviewer 3 Report

Given the existence of so many articles using real data and feature extraction methods this paper does not present applicable advances on to reliable feature extraction methods. The tests with simulated signals are not realistic and are only fit to the methods. From the two real data sets only one performs well! Is it worth applying this method? It would be better to eliminate the tests with simulated signals and add another real case study.

The conclusions do not mention the failed tests with the second data set. Please give some reasoning for this failure.

Details:

Line 58 - It is stated that “In reference [12], an improved VMD method is proposed for the problem that the modal decom-59 position number of VMD needs to be preset.” Please justify and clarify.

Line 85 - It is stated that “However, PSO is inferior to SSA in terms of application range, robustness, 85 and operability.” How come PSO is inferior? Where is the proof, reference? It can be a different approach but, if well applied, PSO is most often superior! Justify or amend.

Lines 264-267: justify the choices of parameters. Give some reasoning.

Line 341 – Figure 5 gives little information to the reader! Perhaps eliminate or remake!

Line 432 – Does it have copyrights? Verify or otherwise reference.

Reviewer 4 Report

This paper proposes a novel fault identification approach based on the vibration signal analysis in the frequency domain. Here are few issues which should be addressed before this manuscript could be recommended for this Journal. 

The literature overview should be expanded. The authors should be aware that vibration signal analysis in the frequency domain is just one of the available techniques for bearing fault diagnosis. Relatively cheap and powerful computational resources available during the last decade did help to develop fault identification techniques in other domains. 

Convolutional neural networks (CNN) and other artificial intelligence assisted algorithms are known to cope very well with different digital image recognition tasks. Therefore, feature extraction techniques producing 2-dimensional digital images from scalar vibration signals have been successfully exploited also in bearing fault diagnosis (when CNN are used for the classification of extracted digital images). A typical example is reported in [A]. The authors should give a wider overview on the current state of the art in bearing fault diagnosis.

Another issue is related to the accuracy of the classification of faults. The accuracy of the detection of different faults should should be carefully discussed. It is also important to discuss (and to demonstrate) how the far presented method is applicable in the presence of noise.  

Finally, the authors should include and discuss comparisons with other alternative methods for bearing fault diagnosis. It is very good that the authors use the Case Western University dataset. That allows straightforward comparisons.  

[A] Permutation entropy based 2D feature extraction for bearing fault diagnosis. Nonlinear Dynamics (2020) vol.102, 1717-1731.  

Round 2

Reviewer 2 Report

The authors made the necessary corrections.